# Bioavailability, Accumulation and Distribution of Toxic Metals (As, Cd, Ni and Pb) and Their Impact on *Sinapis alba* Plant Nutrient Metabolism

**DOI:** 10.3390/ijerph182412947

**Published:** 2021-12-08

**Authors:** Gabriela-Geanina Vasile, Anda-Gabriela Tenea, Cristina Dinu, Ana Maria Mihaela Iordache, Stefania Gheorghe, Mihaela Mureseanu, Luoana Florentina Pascu

**Affiliations:** 1Control Pollution Department, National Research and Development Institute for Industrial Ecology ECOIND, 57-73 Drumul Podu Dambovitei Street, 060652 Bucharest, Romania; gabriela.vasile@incdecoind.ro (G.-G.V.); anda.tenea@incdecoind.ro (A.-G.T.); cristina.dinu@incdecoind.ro (C.D.); luoanapascu@yahoo.com (L.F.P.); 2Chemistry Department, Science Faculty, Craiova University, 107i Bucharest Road, 200585 Craiova, Romania; mihaela_mure@yahoo.com; 3Department of Informatics, Statistics and Mathematics, Romanian—American University, 1B Expozitiei Bld., District 1, 012101 Bucharest, Romania; iordache.ana.maria.mihaela@profesor.rau.ro

**Keywords:** bioaccumulation, translocation, white mustard, trace metals, contaminated soil

## Abstract

This study presents the behavior of white mustard seedlings *Sinapis alba* grown for three months in laboratory polluted soil containing As, Cd, Ni and Pb. Four different experiments were performed in which As was combined with the other three toxic metals in different combinations (As, AsCd, AsCdNi, AsCdNiPb), keeping the same concentrations of As and Cd in all tests and following the national soil quality regulations. The effects of these metals were monitored by the analytical control of metal concentrations in soil and plants, bioavailability tests of mobile metal fractions using three different extracting solutions (DTPA + TEA + CaCl_2_-DTPA, DTPA + CaCl_2_-CAT, and CH_3_COONH_4_ + EDTA-EDTA) and calculation of bioaccumulation and translocation factors. Additionally, micro, and macro-nutrients both in soil and plant (root, stem, leaves, flowers and seeds) were analyzed in order to evaluate the impact of toxic metals on plant nutrient metabolism. Metals were significantly and differently accumulated in the plant tissues, especially under AsCdNi and AsCdNiPb treatments. Significant differences (*p* < 0.05) in the concentration of both As and Cd were highlighted. Translocation could be influenced by the presence of other toxic metals, such as Cd, but also of essential metals, through the competition and antagonism processes existing in plant tissues. Significantly, more Cd and Ni levels were detected in leaves and flowers. Cd was also detected in seeds above the WHO limit, but the results are not statistically significant (*p* > 0.05). The extraction of metallic nutrients (Zn, Cu, Mn, Ni, Mg, K, Fe, Ca, Cr) in the plant was not influenced by the presence of toxic metal combinations, on the contrary, their translocation was more efficient in the aerial parts of the plants. No phytotoxic effects were recorded during the exposure period. The most efficient methods of metal extraction from soil were for As-CAT; Cd-all methods; Pb and Ni-DTPA. The Pearson correlations (*r*) between applied extraction methods and metal detection in plants showed positive correlations for all toxic metals as follows: As-CAT > DTPA > EDTA, Cd-DTPA > CAT > EDTA, Ni-EDTA = DTPA > CAT, Pb-EDTA = DTPA = CAT). The results revealed that *Sinapis alba* has a good ability to accumulate the most bioavailable metals Cd and Ni, to stabilize As at the root level and to block Pb in soil.

## 1. Introduction

Soil pollution is a global problem due to the rapid urbanization and industrialization. Trace metals are the most common type of soil pollutant, being derived from two main sources: natural and anthropogenic [1,2]. The frequent, inappropriate, misinformation and abusive uses and storage of metal preparations or wastes damage the quality of soils used for agricultural purposes. The anthropogenic activities that provide the most significant amounts of metals are improper storage of residual sludge, waste from mining and burning fossil fuels activities, use of fertilizers containing metals, use of amendments for soil fertilization based on biological sludge, industrial and urban waste landfills [3,4,5,6,7]. Soils located in the proximity of areas used for both storage and processing of ores represent a potential risk to plants and animals, due to the excessive accumulation of metals that can be mobilized by leaching and disintegration because of changes in the physical and chemical conditions of the soils [8].

Mining, industrialization, and improper use of fertilizers in the period before the 1990s led to soil contamination with heavy metals in several areas of Romania. Field investigations revealed increased concentrations of Pb (178 to 7466 mg kg^−1^), Cu (26 to 40 mg kg^−1^), Cd (2.5 to 4.6 mg kg^−1^) in the region of Black and White Kőrős-Cris Rivers [9]; Cu (43 to 184 mg kg^−1^), Zn (38 to 161 mg kg^−1^), Ni (25 to 31 mg kg^−1^), Pb (3 to 10 mg kg^−1^) and As (5 to 10 mg kg^−1^) were detected above the allowed limit in the wine-growing area Stefanesti Pietroalele in soil, but also in grapes and wine (Ca > Mg > Fe > Zn > Mn > Pb > Cu > Cr) [10]; in the region Hunedoara-Certej (abandoned mining area), high values of As (16 to 119 mg kg^−1^), Cd (0.53 to 11 mg kg^−1^), Cu (14 to 378 mg kg^−1^), Cr (4 to 73 mg kg^−1^), Ni (3 to 610 mg kg^−1^), Pb (110 to 888 mg kg^−1^), Zn (101 to 2202 mg kg^−1^), and Mn (143 to 3167 mg kg^−1^) have been detected in soil and also in plants (Pb 16 to 32 mg kg^−1^; Ni 4.23 to 10.8 mg kg^−1^; Mn 35 to 60 mg kg^−1^; As 0.14 to 27.7 mg kg^−1^; Cu 1.86 to 15.6 mg kg^−1^; Cr 0.32 to 17 mg kg^−1^) [11].

Metals such as Cu, Zn, Cr, Mn, Ni, Ca, Mg and Fe, are essential for normal plant growth and development, however, the excess of these metals can adversely affect plant growth, photosynthetic and respiratory processes, enzymatic activities, DNA structure and functionality, and membranes integrity [12,13,14]. Non-essential metals such as As, Cd, Hg, Pb fall into the category of pollutants with a potential risk to the environment because they are not biodegradable and are extremely toxic at low concentrations [2,15]. These could cause phytotoxic effects regarding the growth and development of plants, productivity and finally the nutritional quality. The bioavailable metals can be transported through the permeable layers of soil to the groundwater and can be assimilated by the plants, thus entering the food chain [1,16,17,18]. Some studies report As and Cd in plant tissues, and the contents of their bioavailable forms in soil [15,19,20,21,22].

Focusing on trace metals Cd, Ni, Pb and As, we will present below their toxic effects on plants. Cd causes the highest stress to plant tissues. Thus, it negatively affects their morphological and physiological functions by decreasing the absorption and mobility of nutrients in tissues, decreasing biomass production, and significantly reducing the efficiency of the photosynthesis process [23]. Cd in the form of Cd (II) has a chemical similarity to Zn and this inter-substitution can cause malfunctions in metabolic processes [24].

Ni and its compounds can negatively influence the metabolic and physiological processes in plants, leading to imbalances [25]. Ni is a mobile element, being easily absorbed by plants, proportionally with its concentration in soil [26]. In small quantities (0.05 ÷ 10 mg kg^−1^ dry weight, d.w) it is necessary for the growth and development of the plant, being absorbed as ionic form and less as chelates [27]. Ni deficiency leads to chlorosis in young leaves, causing senescence and disrupting nitrogen assimilation and iron absorption. Alternatively, the Ni excess is associated with many side effects, such as reduced germination and plant development, reduced biomass, decreased nutrient absorption, decrease in translocation of most nutrients, necrosis and chlorosis in leaves, and negative effects on the photosynthesis process [25]. The toxic concentration of Ni in the mature leaf tissues is in the range of 10 to 100 mg kg^−1^ d.w. [28].

The major forms of Pb that are released into the soil are Pb oxides, ionic Pb, Pb (II), hydroxides and complexes with Pb oxyanions. Under reducing conditions in soil, Pb sulfides are considered the most stable forms. The accumulation of Pb is limited to leafy vegetables and the surface of the roots [29]. Lead can decrease the absorption and translocation of nutrients into plants, cause oxidative stress and genotoxic effects, inhibit chlorophyll synthesis, and disrupt water balance and membrane integrity [18].

As it is a non-essential and toxic metal in plants, it damages the root development, inhibits root expansion and proliferation, reduces photosynthesis and biomass accumulation, and causes leaf necrosis and suppression in leaf number [26,30,31,32,33]. As can be present in several oxidation states: −III, 0, III, V. As (V) is dominant under aerobic conditions, while As (III) is predominant under reducing conditions. Under extreme reducing conditions, elemental As and AsH_3_ could be present. Arsenite (AsIII) is approximately 100-fold more soluble, mobile, and cytotoxic in nature than arsenate (AsV) [26].

Because heavy metals can cause serious toxic effects on living organisms by accumulation, their incidence and behavior in the environment must be addressed and understood as well as possible, so that control and prevention can be achieved in a way that is as sustainable as possible. In this sense, plants are a “green” way to reduce environmental pollution with metals due to their natural ability to absorb and accumulate metals. The ideal plants used for phytoremediation must meet three essential criteria: to have economic value, to present a low risk after being subjected to contamination, and to have adaptability and tolerance (increased biomass, efficiency in metal absorption) [34]. Medicinal or aromatic plants represent a class that can be used in the phytoremediation of soils contaminated with metals, provided that before being consumed a critical and multidisciplinary analysis of the risks due to contamination is performed. There are still insufficient data on the mechanisms of metal uptake in plants and their translocation in the aerial parts. It is also necessary to experiment and alternate different testing and contamination conditions in order to be able to evaluate the extraction potential of plants and the danger they can present for food quality and public health.

Some plants can block metals in the root, limiting their translocation to aerial tissues. On the contrary, hyper-accumulators translocate and distribute metals both in the root and in aerial organs of plant [35]. Only 0.2% of known plant species can be defined as hyper-accumulators, with metal concentrations of 100–1000 times higher than the average [36]. The studies conducted on different types of medicinal and aromatic plants have indicated that plants have different uptake and accumulation capacities [34,37]. The bioavailability of metals to medicinal/aromatic plants is controlled by several factors associated with the physical-chemical properties of the soil (pH, organic matter content, redox potential, carbonate content, presence of sand), climatic conditions, transfer process and type of metal species, oxidation state, and also the type of plant root [38,39]. The metals in bioavailable forms can provide useful information about the metal concentrations in plant tissues, either bioaccumulated in roots or translocated to the above ground parts of the plant.

The importance of bioaccumulation studies in medicinal and aromatic plants lies in their use in phytomedicine, the composition of food supplements, in food (tea, spices), and in the manufacture of cosmetics (creams, volatile oils, soaps, etc.). Studies have shown, in some situations, the adverse effects of using medicinal plants due to the low-quality raw plant material. Over 50 studies (North America, Western Europe, Australia, India, China, the Middle East) have reported poisoning with metals (such as Al, Cr, As, Hg, Pb and Cd) after the consumption of herbal preparations [29,40]. The World Health Organization (WHO) has imposed maximum allowed values only for three toxic metals, namely Cd (0.3 mg kg^−1^ d.w.), As (1 mg kg^−1^ d.w.) and Pb (10 mg kg^−1^ d.w.) in the medicinal plants which are subsequently used in the preparation of finished products such as juices, essential oils, plant powders [2,41].

The biological model chosen for this study was *Sinapis alba* (white mustard) or *Brassica alba*, a member of dicotyledonate *Brassicaceae*, an aromatic plant or condiment with Mediterranean origins, which is widespread globally and has major economic importance.

This plant has a rapid germination, fast growth, resistance to abiotic stressors and a considerable importance in food and pharmaceutical industries. The mustard seeds present antibacterial, anti-fungal, appetizer, carminative, diaphoretic, digestive, diuretic, emetic, expectorant and stimulant properties. The seeds are frequently used in animal and human food, but the leaves can also be consumed. Mustard cultivation is important for stopping soil erosion and for combating soil pests [41].

For these reasons, our study was designed to investigate the effects of contaminated soil with toxic metals on *Sinapis alba*. The main aim of the study was to evaluate mustard plants grown in soils polluted with toxic metals (above the normal limits), from the seed stage to the mature plant stage, where it bloomed and developed mustard seeds. After three months of metal’s exposure in greenhouse conditions, the concentrations of toxic metals As, Cd, Ni and Pb and of micro and macro-nutrients both in the soil and plant (root, stem, leaves, flowers, and seeds) were analyzed. The bioavailability studies of toxic metals in soil were conducted using three single chemical extraction methods, correlating the values of the mobile metal fraction in soil with total metal content in plant. The bioaccumulation index (BCF), respectively the translocation factor from the root to the aerial parts of the plant (TF) were calculated and statistical hypotheses were issued regarding the impact of toxic metals in the mustard plants subjected to chemical stress conditions.

## 2. Materials and Methods

### 2.1. Analytical Reagents and Certified Reference Materials

Nitric acid 69% and hydrogen peroxide 30%, ultrapure quality (Sigma-Aldrich, Steinheim, Germany) were used for plant tissues digestion. Diethylenetriaminepentaacetic acid (DTPA), Triethanolamine (TEA), Calcium chloride (CaCl_2_), Ethylenediaminetetraacetic acid (EDTA), ammonium acetate (CH_3_COONH_4_) were reagents (analytical grade, Sigma-Aldrich, Germany) used for extraction of metallic mobile fraction. Single element solutions of 10 g L^−1^ As, Cd, Ni, Pb (CPAChem, Bogomilovo, Bulgaria) were used to enrich the soil with metals. Multi-Element Aqueous Certified Reference Material (CRM), type Quality Control Standard 21 (As, Cd, Cu, Cr, Fe, Mn, Ni, Pb and Zn 100 mg L^−1^, LGC quality, Wesel, Germany) was used for calibration curve for metals detection. For Ca, Mg, Na, K detection a Multi-Element solution was prepared using 10 g L^−1^ unielement CRMs (CPAChem).

The quality control of the results (metals in soil and plant samples) was performed with the following matrix type CRMs: SQCI-001 (Metals in Soil, NSI Lab Solution, Raleigh, NC, USA), BCR-483 (Sewage sludge amended soil, Joint Research Centre, Geel, Belgium), BCR-482 (Lichen, Joint Research Centre, Brussels, Belgium), NIST 1515 (Apple Leaves, National Institute of Standards and Technology, Gaithersburg, MD, USA), NIST 1573a (Tomato Leaves, National Institute of Standards and Technology, Gaithersburg, MD, USA).

### 2.2. Equipment

ICP-EOS AVIO 500 Perkin Elmer Spectrometer (Waltham, MA, USA) was used for simultaneous detection of the metals. The pretreatment process of the soils was performed using a grinding mill Retsch RM 100 (Haan, Germany), a sieving system Analysette 3 Spartan Fritsch (Idar, Oberstein), and an Ethos Up Milestone Microwave System (Sorisole, Italy). The plant tissues were dried at 50 °C in a Memmert oven UF 110 (Schwabach, Germany) and digestion process was performed in the abovementioned Microwave System.

### 2.3. Experimental Design and Plant Materials

The experiments were performed in a greenhouse of 6 m^2^ (Gothic model) with vertical side walls made of 4 mm thick polycarbonate and Al structure, with a sliding door and two manually folding skylights.

The *Sinapis a**lba* seeds were provided by MicroBioTests Belgium—SIA 020,719 with a minimum of 70% guaranteed germination in the negative controls after 3 days of incubation. A universal soil type amendment for plant culture, containing a mixture of peat from decomposed swamps, wood fibers, green compost, tree bark humus, nitrogen—phosphorus –potassium fertilizer from a local producer was used. According to the producer, the universal soil contained 50–400 mg L^−1^ of N, 50–200 mg L^−1^ of P_2_O_5_, 50–200 mg L^−1^ of K_2_O, KCl less than 3 g L^−1^, minimum 67% organic content, pH 6.5 ± 0.5, humidity 60%, without toxic chemicals.

Separately, garden soil harvested from a depth of 0–50 cm was used. The soil and the amendment were shredded and the wood parts, the vegetation and the stones were removed. Both solid materials were air dried at room temperature for 14 days and sieved through a sieve with a mesh size less than 5 mm, in order to homogenize the entire quantity. The amendment was mixed with garden soil in a ratio of 1:3.

The soil and amendment mixture, as well as the mustard seeds used in the experimental studies were analyzed for metal content. After drying the soil samples at air temperature, the fraction less than 150 µm was selected, and the metals were extracted from about 1 g of soil using aqua regia mixture (9 mL HCl and 3 mL HNO_3_) in a microwave system. In addition, 1 g of mustard seed powder was mixed with 9 mL of HNO_3_ and 1 mL of H_2_O_2_, heated until complete digestion with a special program for plant tissue. After digestion process, all the solutions were filtered and brought with ultra-pure water to volumetric flasks of 50 mL for soil solutions and 25 mL for vegetal extracts.

The determination of the metal content (As, Cd, Ni, Pb) was performed both from soil and plants. Three individual soil samples were collected at the begging of the experiments in order to control the metal content.

The levels of As, Cd, Cu, Co, Cr, Fe, Mn, Ni, Pb, Zn, Ca, Mg, Na, K were measured both in soils and tissue parts. Other analyzed parameters were pH, conductivity, total nitrogen, total phosphorus, humus and total carbon, chlorides, sulphates, bicarbonates, organochlorine, triazine and phosphoric pesticides.

Five different treatments were set-up. The selected metals, tested concentrations, number of test replicates and number of plants per treatment are presented in Table 1.

The control soil was divided into five parts, one part remaining the control sample and the other lots were enriched with metals. Each treatment was conducted in two replicates. Each batch of soil was sprayed with metal-enriched tap water solution and left to stabilize for three weeks before mustard seedlings planting.

The metal concentrations were selected to be either at the alert threshold for sensitive use or above it, but not exceeding the intervention limits according to Romanian legislation (Table 2). The selected concentrations for Ni in T3 treatment was above the alert value (75 mg kg^−1^) and below the intervention limit for land with sensitive use (150 mg kg^−1^), simulating a polluted soil specific to a mining area in Romania [11].

The seedlings were grown in the uncontaminated amended soil until they reached 3 cm in size and then planted in identical plastic pots (30 cm^3^ volume, 5 kg of contaminated soil each) and placed in the greenhouse. The study was conducted over o period of 3 months, from May to August, until the mustard seedlings reached maturity; bloomed and developed sheath seeds. In the greenhouse, the average temperature during the entire period was around 26 ± 4 °C (minimum 13.5 °C at night and maximum 33.5 °C during the day), atmospheric air humidity was in the range 52% to 63%, natural light of 6500 lux in rainy weather, 12,500 lux in cloudy weather and around 31,800 lux in sunny weather. Watering was carried out twice per day to keep a constant humidity in soil of about 60% from the maximum moisture retention capacity of used soil.

### 2.4. Metallic Mobile Fraction Evaluation Procedures

The bioavailable metal fraction in soil (control and polluted) was evaluated with three different chemical extraction procedures. A detailed description of the applied methods is presented in Table 3.

### 2.5. Statistical Analysis

The results of metal concentrations in the control and polluted soils before starting the experiments were expressed as average (*n* = 3) ± expanded uncertainty (ue) with 95% confidence level using a coverage factor of k = 2. The results of metal concentrations were expressed as average ± standard deviation (SD) with *n = 3* for soil samples in mg kg^−1^ d.w.

Regarding the plants, three specimens of each replicate were harvested when they reached flowering, separated into organs and analyzed. Thus, six different plants were analyzed for the same treatment (plants, *n = 6,* in mg kg^−1^ d.w.). The other two plants that remained in each pot for a specific treatment were left to reach maturity for seed harvesting. In this case, the seeds were harvested, and three different samples were analyzed for each pot.

Two-way ANOVA with Tukey’s HSD post hoc test was used to determine differences among experiments. Statistical analysis of data was performed using SAS Enterprise Guide. The values with the same letter are not significant different (*p* < 0.05) according to Tukey HSD.

F-Test (two samples for variances) was used for significant differences (*p* < 0.05) between the results of mobile metals obtained with chemical extraction procedures.

In addition, the Pearson correlation (*r*) was used for correlations between concentrations of metals in the soil mobile fraction and total values of metal concentrations extracted by the plants.

### 2.6. Data Analyses

The bioaccumulation factor (BCF) and the translocation factor (TF) were calculated in order to evaluate the plant’s capacity to accumulate metals from soil and to transfer them from the root to the aerial parts. The root bioaccumulation factor (BCF) was calculated as the ratio between the concentration of the metal (Me) in the plant root and the initial concentration of the element in the soil [46]:(1)$$\mathrm{BCF}=\frac{\mathrm{Me}\;\mathrm{concentration}\;\mathrm{in}\;\mathrm{root}}{\mathrm{Me}\;\mathrm{concentration}\;\mathrm{in}\;\mathrm{soil}\;}\;$$

The translocation factor (TF) was calculated as the ratio of metal concentrations in the aerial part of the plant to those in the roots, indicating the plant’s ability to translocate metals from roots to shoots [47]:(2)$$\mathrm{TF}=\frac{\mathrm{Me}\;\mathrm{concentration}\;\mathrm{in}\;\mathrm{aerian}\;\mathrm{tissue}}{\mathrm{Me}\;\mathrm{concentration}\;\mathrm{in}\;\mathrm{root}\;}\;$$

The BCF and TF values were calculated for toxic metals (As, Cd, Ni, and Pb), micro-nutrients (Cu, Cr, Mn, Zn) and macro-nutrients (Ca, Mg, Fe, K) in all experiments. An average value of three different samples for soil and six different samples for plant tissue (three plants from each replicate, such as T1-1 and T1-2) was used. Olowoyo et al. stated that a BCF value higher than 1 suggests metals accumulation, a BCF value around 1 shows that the plant was not influenced by the metal and BCF less than 1 indicates no metal uptake [48].

The TF value higher than 1 indicates that the plants effectively translocate metals from root to the above ground plant parts [48].

The results were correlated and compared with control sample values and also with the reference values for the soil and plant quality [41,42].

## 3. Results

### 3.1. Chemical Analysis of Soils and Seeds

The results of physical-chemical parameters determined in the control soil showed a conductivity of 575 μS cm^−1^, TOC 6.21%, K 3056 mg kg^−1^ d.w., N_total_ 1.41%, C_total_ 15.3%, P_total_ 1550 mg kg^−1^ d.w., 514 mg kg^−1^ d.w. chloride, 150 mg kg^−1^ d.w. sulphate, and 410 mg kg^−1^ d.w. bicarbonates. Organochlorine, triazine and phosphoric pesticides were not present in the control and contaminated soils.

The pH value for control samples was 6.94 and the pH values for contaminated soil samples ranged from 6.85 to 7.21 pH units. The pH value indicated a neutral reaction of the control soil and a weak acid reaction of the polluted soils.

Due to nitrogen and phosphorus content, the soil was considered a clay soil, rich in organic matter. The C/N ratio of 12 indicated a good mineralization reaction in the soil and the release of nitrogen, which is available for plant uptake.

The results of toxic metal concentrations are presented in Table 4. Moreover, elements such as Ca, Mg, Na, Fe, K, Cu, Co, Cr, Mn, Zn considered as essential for plant growth (in the appropriate concentrations) were analyzed.

Tap water used in the experimental tests did not contain the metals of interest, namely As, Cd, Ni and Pb or other toxic metals. Moreover, Ca 42.8 µg L^−1^, Fe 37.3 µg L^−1^, Zn 14.3 µg L^−1^, Mg 3.6 µg L^−1^, Cu 5.4 µg L^−1^, Mn 2.8 µg L^−1^, Al 111 µg L^−1^ were found in the soaking water. The results represent the mean value of ten water samples, analyzed over the entire period of the experiments.

### 3.2. Bioavailability Tests

Bioavailability tests were performed using three different single chemical extraction procedures, as shown in Table 3.

All the results regarding metal mobile fraction versus total content in control and polluted treatments were plotted in Figure 1, Figure 2 and Figure 3. If the values of mobile metals were compared for the same treatment, it was noted that the highest results for Cd and Pb were obtained with the DTPA method, but without significant differences between methods (*p* values 0.17 to 0.48) (Figure 1B,D, Table 5). For Ni, DTPA method obtained significant results compared with EDTA (*p* = 0.007) and CAT (*p* = 0.0103), Figure 1C. Only for As mobile fraction the best results were obtained with CAT method (*p*-values = 0.014; 0.0004) (Figure 1A, Table 5).

If no metal addition was performed (Cd in T1; Ni in T1 and T2, respectively Pb in T1, T2 and T3), no differences were observed between the concentrations of mobile metals.

Although the bioavailability tests indicated that Pb was found in a high proportion in mobile form (between 50% and 100% depending on the applied method), the plants extracted only a small amount of Pb, which was bounded in the roots.

For the nutrients needed in the growth processes, i.e., Zn, Ca, K, Mn, Fe, more differences between the results obtained with different extraction methods were highlighted. For example, EDTA method extracted significantly more Zn (*p* = 0.0006) (Figure 2A), Ca (*p* = 0.016) (Figure 3C), and Mn (*p* = 0.005) than the other methods (Table 5). DTPA extracted K (*p* < 0.001) (Figure 3A). The CAT method was suitable for mobile Fe (*p* = 0.0006), Figure 3D.

The values of mobile Fe (Figure 3D) were not compared with total content of Fe for each control and treatment, due to the large difference between total and mobile concentration (see Fe total content in Table 3). No graphic data were reported for Cr, because none of the applied methods extracted mobile Cr.

### 3.3. Metal Concentration in Plant Tissues after Exposure Statistical Analyses

No phytotoxic effects were observed in mustard plants during the exposure period. The plants were vigorous with normal biomass without evidence of chlorosis or leaf loss. In the visual analysis of the plants, small differences were observed in the height and thickness of the stems and the abundance of the inflorescences. In the T2 treatment, richer inflorescences were observed, and the stems were taller and thinner compared to the control plants and the plants exposed in the T3 and T4 treatments.

Metals accumulation in different tissues (root, stem, leaves, flower, sheath, seeds) and total accumulation in *S. alba* plants over a period of three months is presented in Table 6, Table 7 and Table 8. The results revealed concentrations of As in root, Cd in leaves, flower and sheaths and Ni in all plants part. Additionally, Cu, Zn and Fe were detected in roots, leaves, flowers and sheaths. Cu and Zn concentrations exceeded the limit values for plant developments, but no phytotoxic effects were observed. The As concentrations in seeds were less than 1 mg kg^−1^ d.w., while for Cd exceeded 0.3 mg kg^−1^ d.w., the WHO limit set for medicinal plants. The experiments started with 0.29 mg kg^−1^ Cd in mustard seeds. If we take into consideration the expanded uncertainty of the mean result (0.03 mg kg^−1^), the WHO limit for Cd was reached. More than 1 mgkg^−1^ d.w. Cd concentration was detected in seeds, especially in the experiments T2 (1.16 ± 0.10 mg kg^−1^), T3 (1.75 ± 0.16 mg kg^−1^) and T4 (1.26 ± 0.11 mg kg^−1^), without significant differences in concentrations between treatments.

In order to highlight the significant differences between the four treatments (T1, T2, T3 and T4) and the detected metal concentrations in plant tissues, Tukey HSD Test (*p* < 0.05) was applied.

Only treatments in which the pollutant was present either alone or in a mixture with other metals as a source of soil pollution were considered for statistical analysis. No statistical analysis was performed for Pb because this element was not found to be bioaccumulated in any plant tissue. For As, statistical analysis was performed only for the tissues in which As is bioaccumulated.

Statistical data showed significant differences (*p* < 0.05) in toxic metal concentration detected in plants (Table 6) as follows: (*i*) As was found in significant quantities in the roots in all experiments but especially in T3 and T4. There are no significant differences between T1 and T2 treatments; (*ii*) Cd was found to be significant in flowers (T4) and leaves (T3); (*iii*) Ni was present in significantly higher concentrations in flowers and seeds (T3, T4).

The Zn, Cu, Cr, Mn micronutrients (Table 7), respectively, the Ca, Mg, K, Fe macro nutrients (Table 8) showed different levels of accumulation depending on the applied treatment. It has been found that Ca and Mg macro nutrients accumulated significantly especially in leaves, roots and seeds in all treatments compared to control. K has become apparent in stems and leaves, especially after the T1 treatment. The applied treatments determined a lower concentration of Fe at the level of roots and an increased one at the level of leaves and seeds compared to the control tests. Cu, Cr and Zn although not added in experiments were significantly present in flowers/roots, leaves/flowers and in sheaths, respectively, after T4 treatment compared to other treatments (T1, T2 and T3) and control. There were no significant changes in Mn accumulation, except for the T1 treatment at the level of leaves where it was found twice as much compared to the control.

### 3.4. Bioaccumulation Index (BCF)

The BCF index values were calculated using the average value of three independent determinations for each type of soil and six determinations for plant tissue. The bioaccumulation factors indicated that only Cd and Zn were accumulated in the mustard roots, while the other tested metals, both toxic and essentials, had BCF values lower than 1 (Figure 4).

Regarding Cd, the highest BCF values were obtained in control and T1, where no Cd was added. In the experiments with Cd addition (T2, T3, T4), bioaccumulation in the mustard root was observed for T4 treatment (BCF = 1.25). The same pattern was identified for Zn, bioaccumulation in the root was reported in control, T1 and T4 treatments. Regarding K, high values of BCF (higher than 10) were recorded and for this reason the values were not entered in Figure 5. K accumulated in root both for control (BCF = 10) and treatments, ranging from 18.5 (T1) to 11.7 (T2), 10.7 (T3), and 12.8 (T4), respectively.

No significant differences were reported between control and treatments for all analyzed metals (*p*-value > 0.05).

### 3.5. Translocation of Metals (TF) in Plant Tissues

The TF index values were calculated using the average value of six independent determinations for each type of plant tissue. Regarding As, the TF index had the highest value in T4 treatment compared to the other treatments and an As accumulation in the leaves was reported (TF = 2.2). In addition, the bioaccumulation of As occurred in sheaths (T2, TF_sheaths/root_ = 1.4), stem (T4) and leaves (T3, T4) (Figure 5A). The mustard seeds, despite the applied treatment, did not accumulate As.

In the treatments with Cd addition (T2, T3, T4) it was observed that Cd was accumulated in various parts of the plant, either in stem (T3, T4), leaves (T2, T3, T4), flowers (T3), or even in sheath (T2, T3). The only part of the plant in which Cd did not accumulate and the recorded concentration was situated at the normal level was the seed, which represents the edible part of the plant (Figure 5B). The highest TF value (TF = 3.25) was recorded in T3 leaves, but the average concentration of 7.2 mg kg^−1^ was below the phytotoxic value of Cd in plants (10 mg kg^−1^) [49]. The highest concentration of Cd was reported in leaves under T4 treatment (7.6 mg kg^−1^).

Regarding Ni, it was observed that Ni accumulated in flowers and mustard seeds in both tests with Ni addition (T3, T4), (Figure 5C). The TF values indicate a higher accumulation in flowers than in the seeds, the average concentrations recorded in flowers (T3 = 26.7 mg kg^−1^; T4 = 39.2 mg kg^−1^) being close to or even exceeding the phytotoxic value (30 mg kg^−1^). In the mustard seeds were recorded Ni values in the range of 15 mg kg^−1^ to 20 mg kg^−1^, much higher concentrations than the normal range of values (1 mg kg^−1^ ÷ 5 mg kg^−1^) [49].

Pb was retained in soil, being extracted by the mustard roots from the contaminated soils in a very low concentration (about 2 mg kg^−1^). Even in the Pb-contaminated test (T4), Pb was found only in roots, all other parts of the plant remaining unaffected by the lead.

For the metals used by the plant in different biochemical processes (Cr, Mn, Zn), the concentrations recorded in the contaminated treatments were in the same range as control samples, bioaccumulation was observed in the plant tissues. In the control plants, Zn did not accumulate in any part of the white mustard, while in the T1 treatment, Zn accumulated in leaves (TF = 1.5) and flowers (TF = 1.3), Figure 6A. In T2 treatment, Zn accumulated in leaves, flowers and mustard seeds, the highest TF value was recorded in the seeds (TF = 1.42). In T3 treatment, Zn accumulated in flowers (TF = 2.25) and mustard seeds (TF = 1.76), while in T4 treatment, Zn accumulated in leaves (TF = 1.73) and mustard flowers (TF = 1.25). The highest concentration absorbed by the plants was founded in the leaves from the T4 treatment (187 mg kg^−1^), a value situated below the phytotoxic concentration (200 mg kg^−1^) [49].

Regarding Cr, the experimental data indicated its accumulation in the stem and leaves from the T4 treatment (TF = 2.92), as well as in the sheaths from the T3 treatment (TF = 3.72), Figure 6B. The average value recorded in the T3 mustard stem (3.54 mg kg^−1^) exceeded the phytotoxic value in plant tissue (2 mg kg^−1^ Cr) [49].

The highest value of Cu was recorded in T4 mustard flowers (45 mg kg^−1^), a value of 2.5 times above the phytotoxic concentration (20 mg kg^−1^) [49], TF index being 3.23, Figure 6C. However, the value determined in seeds from the same experiment (7.5 mg kg^−1^) was in the normal range of concentrations (3 ÷ 15 mg kg^−1^) [49].

Mn has accumulated mainly in the leaves (TF_T1_ = 1.48; TF_T2_ = 1.75; TF_T3_ = 1.05) and less in the seeds (TF_T2_ = 1.24), Figure 6D. In T1, T2 and T3 tests, Mn values varied in the range 22 ÷ 34 mg kg^−1^. The highest amount of Mn in a tissue in all tests including control was founded in T4 roots (58 mg kg^−1^).

Ca and Mg accumulated mainly in the leaves, with BCF values between 4.91 (T2) and 5.83 (T1) for Ca, respectively, 3.55 (T2) and 4.50 (T1) for Mg. Ca it was also accumulated in sheaths, at TF values between 2.3 (T2) and 3.64 (T3) and less in flowers and seeds (Figure 7A). In contrast, Mg was accumulated more in seeds than in sheaths, with TF values varying between 2.0 (T3) and 2.82 (T2) in seeds (Figure 7B). With few exceptions, K was accumulated mainly in strain, both for control and treatments (Figure 7C). Regarding Fe accumulation, TF values were well below 1 in both control and treatments. Thus, the maximum values recorded were 0.08 in stem (T4), 0.17 in leaves (T3), 0.34 in flowers (T4), 0.18 in sheaths (T3), respectively 0.24 in seeds (T3).

## 4. Discussion

### 4.1. Metal Detection in Soil and Plants

The metals behavior in the contaminated soil and their bioavailability to *S. alba* were assessed using BCF index, TF index and three single chemical extraction procedures. Mustard seeds used in the treatments did not contain toxic metals above the WHO limits. However, a concentration of 0.29 mg kg^−1^ d.w. was determinate for Cd which recorded a value at the normative limit.

The initial contaminated soil characterization showed As, Cd, Ni and Pb values situated between alert threshold and intervention threshold limits according to Romanian Order for soil quality [42]. For the Cu, Co, Cr, Fe, Mn, Zn, Ca, Mg, Na and K metals, normal values were registered. The performed experiments reproduced in the laboratory the soils contaminated with metals, similar to those of some polluted regions in Romania which were presented in the introduction part [9,10,11]. The contaminated soils used in the experiments contained plant growth amendments such as TOC, P, N, sulphates, bicarbonates, and essential metals. The literature reported that these amendments could contribute to the reduction in the degree of toxicity of metals on mustard plants [50]. In addition, soil properties such as pH or organic carbon content have strong effects on soil solubility and speciation of the metals. The mobility and availability of metals is low in soil with high pH, clay and organic matter content and these factors contribute to the poor As and Pb bioavailability [51]. This information may explain the normal development of plants in the presence of toxic metals, the nutrient matrix and soil characteristics contributing to the decrease in the toxicity of the studied metals.

As, as a single pollutant or in combination with Cd, Ni and Pb, had the same concentration in all treatments (about 15 mg kg^−1^ d.w.). After three months of exposure, some effects were observed. As was immobilized in the plant roots and was presented in leaves and sheath. The As concentration in plants was up to a normal value without phytotoxic effects (5 mg kg^−1^) and the WHO limit (1 mg kg^−1^) was reached in plants in the range of 4 mgkg^−1^ d.w. to 10 mg kg^−1^ d.w., depending on the treatment, even if the initial contamination was the same. The observed effects were probably induced by the presence of other metals. The highest accumulation was estimated in the T3 experiment were As was combined with Cd and Ni in contaminated soil. Studies of metal contamination of ruderal vegetation in areas adjacent to mining have shown that the accumulation of As in plants is in the 0.14 to 27 mg kg^−1^ concentration range [11], which confirms the tolerance of plants to As.

Pb was blocked in the soil and very low concentration were registered at root level. Studies performed on various plant species, including *Brassica* species, showed that Pb has a low bioavailability in the soil and that it requires certain supplements such as EDTA to be available to plants. Pb can accumulate in roots and less in stems and leaves [52]. Some plants tolerate high concentration of Pb in soil (>1000 mg kg^−1^) such as *Cyamopsis tetragonoloba* (guar) or *Sesamum indicum* L. (sesame) [51].

Cd was present in flowers, leaves and sheath, especially in the case of T3 and T4 experiments where As was combined with Cd, Ni and Pb. The literature reports a competition process between metals regarding their takeover by the plant and their translocation in the upper parts [53]. Soil contamination studies with metals in areas adjacent to the mining complexes in Baia Mare, Romania, have revealed Cd concentrations of 2–3 mg kg^−1^ in the grape leaves, starting from a pollution with 15.84 ± 1.36 mg kg^−1^ Cd in soil [54], while our studies showed values of 3 to 10 mg kg^−1^ in *S. alba* starting from a value of Cd contamination in soil of about 3 mg kg^−1^, indicating a good phytoextraction capacity of Cd.

In addition, the metal micro- and macronutrients such as Ni, Cu, Zn, Mn, C, Mg, K, Fe, Cr were present in roots, flower, leaves and seeds. This observation could be correlated with the physiological need of the plant for essential metals in the different development stages and in various physiological processes such as photosynthesis, biomass production, mechanical strength, synthesis, and activation of enzymes, etc. [55].

If we take into consideration the total concentration of metals detected in mustard plants, the studied metals exceed the WHO limits established for As (1 mg kg^−1^) and Cd (0.3 mg kg^−1^) and also the normal values presented in literature for plants: 5 mg kg^−1^ (As), <0.1–1 mg kg^−1^ (Cd). Ni exceeds the normal value in plants of 0.1–5 mgkg^−1^. These elements also exceed the phytotoxic values of 10 mg kg^−1^ for Cd and 30 mg kg^−1^ for Ni [53].

Even if the metal concentration exceeds the phytotoxic concentration in plants, we showed that in combination, the metals have no phytotoxic effects. Furthermore, the plant has grown to blooming and to seeds production. The literature data shows various studies where the plants exposed to metals combinations resist and tolerate this stress; non-significant changes on biomass and phytotoxic effects were observed [53,56,57,58].

The mustard seeds are the edible part used in medicinal, food and cosmetic products. The presence of toxic metals could have an influence on the balance of essential metals from the mustard seed, the most used part of mustard plants. The seeds contain a variety of minerals, including Fe, Mg, Zn, Ca, and P [59]. As, Cd, Ni and Pb in seeds were in lower concentration compared to the other plant organs and controls. Cd exceeds the WHO limit in seeds, the results that could be alarming for seed consumption, if the mustard plants are cultivated in Cd polluted soils.

We observed that *S. alba* had the capacity to select at seed level the essential elements such as Cu (6.49 mg kg^−1^ in T3 to 9.63 mg kg^−1^ in T2), Ni (12.78 mg kg^−1^ in T4 and 20.85 mg kg^−1^ in T3) and Zn (67 mg kg^−1^ in T3 to 94 mg kg^−1^ in T2). *Brassica napus* and *Raphanus sativus* were moderately tolerant when grown on a multi-metal contaminated soil (Cd, Cr, Cu, Ni, Pb, and Zn) [60]. Generally, the toxic metals are not bioaccumulated by the plants because these have developed tolerance or adaptation biochemical mechanisms, which prevent or block their extraction from the soil [52,61,62]. The plants benefit from a system of tolerance and sensitivity to stressors, which is difficult to be monitored. This behavior could be explained by the presence of some specific metals transporters that differentiate toxic metals from essential ones. The protein transporters have the role of translocation or blocking metals or other types of substances in the membranes. Their functioning is essential in the process of detoxification or tolerance [63]. The metals with known biological role for plant development such as Cu, Zn, Ni, Mg, Fe, Cr can be efficiently extracted by plant and depending on their accumulated concentrations they can become dangerous for both plants and consumers. For example, Zn is an essential micro element for plants, but it can become toxic to mature plants at higher concentrations (between 100 and 400 mg kg^−1^) [28]. Our results indicated total Zn values in the range of 400 to 600 mg kg^−1^ (higher in the case of T4 and T1 treatments) without showing phytotoxic effects, which confirm a resistance of *S. alba* plants to high concentrations of Zn.

Regarding the metal accumulation, these could be influenced by the presence of other toxic or essential metals and nutrients such as P, N and organic maters. For example, some authors correlated that plant-available As in the soil and its uptake in the plant increased with increasing of P concentration in the soil due to competition between arsenate and P [40]. Instead, Cd bioavailability decreased with the increasing of P or sulphates concentrations in soil [40,64]. The concentration of P_total_ in the contaminated soil was initially determinated at 1550 mg kg^−1^, this value indicating the possibility of a higher bioavailability of Cd compared to As.

The ANOVA statistical analysis showed the metallic elements that were significantly accumulated in plants tissues. Significant differences for both the concentration of As and Cd were highlighted, at a significance *p* < 0.05.

The As did not pose problems in aerial parts of plants, but only at the root level, and its accumulation could be influenced by Ni and Pb from T3 and T4 experiments, where As was more extracted by plants roots. No significant concentrations of As and Pb were observed for all experiments compared to control. Significantly more Cd and Zn reached the leaves (T3 and T4), *p* < 0.05. In addition, Cd, Ni, and Zn were found in flowers and Cr in seeds and sheaths, all in T4 treatments. The results were comparable with other studies on *Populus* spp., *Brassica* spp., *Mentha* spp., *Ocinum* spp., *Atriplex halimus* [52,53,58,65].

No toxic metals (As, Cd, Pb) were translocated significantly in seeds excepting the total content of Ni that exceeded the normal value in plants (0.1–5 mg kg^−1^) and the phytotoxic limit (30 mg kg^−1^) [53]. Even the Cd concentration detected in the seeds exceeded the WHO limit (0.3 mg kg^−1^) and the BCF was higher than 1 (because mustard plants accumulated Cd in root), the TF value was less than 1 (but higher than 0.5). Therefore, Cd translocation from roots in aerial parts, respectively in seeds was not efficient; the statistical analyses showed that Cd accumulation at seeds levels was not significant.

### 4.2. Bioavailability Tests

EDTA and DTPA are the most used procedures for mobile toxic metal extraction from soil, especially in soil phytoremediation studies [66,67,68].

Cd was efficiently extracted from soil using all three procedures, especially in the T3 experiment. The efficiency of the extraction procedure was influenced by the experimental design, respectively by the metal combinations and their concentrations. For example, mobile As had a good extraction in T1 and T2 using the CAT extraction method while in the T3 and T4, the efficiency of the extraction procedure was much lower. If the concentrations of mobile metals exceed the binding capacity of the used extraction solutions, the quantities of metals in mobile form decreased and thus may not reflect the actual behavior and bioavailability. This can explain the decrease in the As concentration obtained with the CAT method in T3 and T4 treatments, where Ni was added [69,70,71].

The positive correlations between mobile metal concentrations from soil and total content extracted by the plants have been shown for all metals except Mn.

Pearson correlation (*r*) revealed a strong correlation (*r* = 0.96; 0.92; 0.66) between the total content of Cd in plants and mobile Cd in soils (DTPA > CAT > EDTA). No significant differences were reported between methods, the conclusion being reported also by other studies [72].

Ni showed *r* = 0.98; 0.98; 0.66, a good correlation for EDTA = DTPA, that was higher than CAT.

As registered positive correlation for EDTA method (*r* = 0.88) and less for DTPA (*r* = 0.51) and CAT (*r* = 0.44).

For Pb the correlations were positive, but lower than 0.5 (*r* = 0.49) for all methods.

Good correlations were obtained also for some essential metals such as Zn (*r* = 0.86 − CAT), Cu (*r* = 0.60 − DTPA), Ca (*r* = 0.94 − EDTA), K (*r* = 0.87 − CAT), Mg (*r* = 0.52 − DTPA). Negative correlations were registered for Mn (*r* = −0.25; −0.35; −0.50) for all applied chemical extraction procedures.

### 4.3. Bioaccumulation and Translocation of Metals in Plants

The plant’s ability to accumulate metals from soil was evaluated based on the bioaccumulation factor (BCF). The BCF indexes for toxic metals indicated that only for Cd the BCF limit (value higher than 1) was exceeded. As, Ni and Pb were not efficiently absorbed from soil in plant roots. Regarding the micronutrients, the BCF index recorded a value higher than 1 for Zn.

The translocation of the metallic elements from roots to the aerial parts of the plant was evaluated by the translocation index (TF). The translocation data showed that the transfer of both toxic and trace metals was done differently, being influenced by several factors such as antagonism and competition between elements, the stage of plant development, the type of organ, and its physiological functions. Analyzing the behavior of the plant in the presence of As, we noticed that As can be efficiently translocated (TF > 1) from roots to leaves and sheaths. The presence of other toxic metallic elements (T2 to T4) may influence the As translocation. Cd and Ni showed TF > 1, especially in the experiments in which they were added. There was an efficiency of Cd translocation from roots to sheaths, and also in leaves and flowers. The highest concentrations of Cd were determined also by other authors in *S. alba* plants exposed to toxic metals [73]. According to the literature, *S. alba* could be considered a good phytoextractor of Cd (BCF and TF > 1), not suitable for extraction of Pb (BCF and TF < 1) and good translocator of Cd, As, and Ni [51]. Shukla and Behera presented the mustard (*Brassica campestris* L.) as a hyper-accumulator of Cu, Ni, Pb, Zn, the accumulation ratios being 2.43 (Ni), 2.37 (Pb), 1.47 (Zn), 1.20 (Cu), respectively 0.73 (Cd) [74]. The tests performed in this study indicated a bioaccumulation of Ni and Zn in various aerial parts of mustard plants, and also reported Cd as a potential risk factor if it occurs in the soil in bioavailable form. The experimental tests did not show an influence of Pb, and Cu concentration was low.

Due to the physiological function of some metals such as Ca, Mg, Ni, Zn, Mn, and Cu, they have a higher translocation efficiency in leaves, flowers, and seeds. It can be appreciated that the translocation of Zn in mustard seeds can be negatively influenced by the presence of As (T1), while in experiments in which As was combined with Cd, Ni, and Pb, the Zn translocation was more efficient. Study on the assessment of heavy metals tolerance of plants grown in contaminated urban soil showed that *S. lycopersicum* and *B. juncea* had the ability to transport heavy metals from roots to shoots, especially for Zn [75]. In addition, Cd in the form of Cd (II) has a chemical similarity to Zn [24] which leads to a competitive phytoextraction effect between Cd and Zn.

It was also observed that the translocation of Mn in leaves and seeds is efficient in T1÷T3 experiments, but not in T4 where Pb was added. These findings indicate that white mustard plants have complex detoxification mechanisms that can be influenced by the type of metal and its concentration. It is very important to specify that all toxic metals (As, Cd, Pb) except Ni were not efficiently translocated at the seeds level, but they had higher translocation values than in the controls.

## 5. Conclusions

This study monitored the effects of toxic metals on mustard plants’ growth and development. The mustard proved to be very resistant, as it was developed, flourished, and made sheaths with seeds in all tests regardless of the mixture of metals. The analysis of plant organs revealed As in roots, Cd and Ni in leaves, flowers and seeds. Statistical differences were observed for both As and Cd. Pb was not detected, either in root or in the aerial parts.

The bioavailability studies regarding the toxic metals in the soil were conducted using three single chemical extraction procedures. The tests indicated the appropriate method for each toxic metal, as follows: As − CAT, Cd − DTPA > CAT > EDTA, Pb and Ni − DTPA. Pearson correlations (*r*) between applied extraction methods and metal detection in plants showed positive correlations for toxic metals as follows: As − EDTA > DTPA > CAT, Cd − DTPA > CAT > EDTA, Ni − EDTA = DTPA > CAT, Pb − EDTA = DTPA = CAT. Cd and Ni proved to be much more mobile than As and Pb.

The bioaccumulation index (BCF) had values higher than 1 for Cd, Zn and K. The translocation factor (TF) from the root to the aerial parts of the plant showed higher value than 1 for all metallic elements: As, Cd, Zn, Mn in leaves; Cd, Ni, Zn, Cu in flowers; As, Cd, Cu in sheaths; Ni, Zn, Mn in seeds.

The results revealed that *S. alba* plants have a good ability to accumulate Cd and Ni, retains As at the root level, and stabilizes Pb in soil. Due to its economic value in food production, it is recommended to cultivate S. *alba* in soils with low Cd and Ni content.

## Figures and Tables

**Figure 1 ijerph-18-12947-f001:**
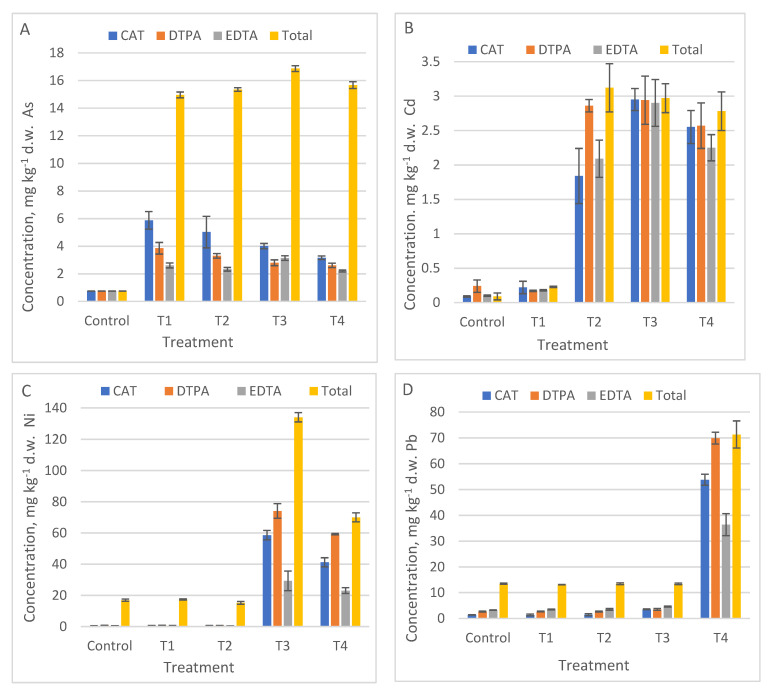
The mobile metallic concentration in Control, and T1 to T4 treatments using three single chemical extraction procedures: DTPA, CAT, EDTA compared to total content, (average ± SD, *n* = 3): (**A**) As, (**B**) Cd, (**C**) Ni, (**D**) Pb.

**Figure 2 ijerph-18-12947-f002:**
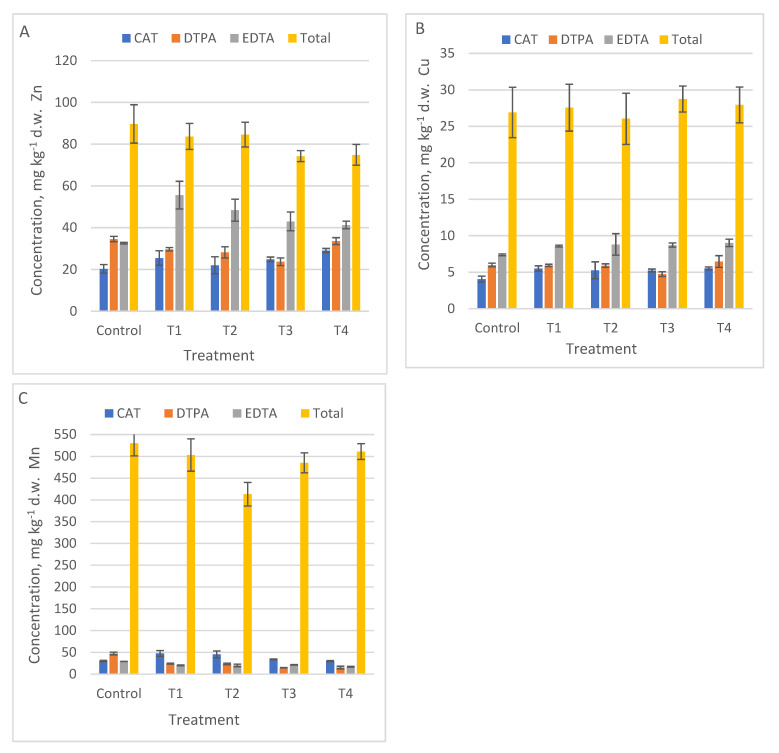
The mobile metallic concentration of micronutrients in Control, and T1 to T4 treatments using three single chemical extraction procedures: DTPA, CAT, EDTA compared to total content, (average ± SD, *n* = 3): (**A**) Zn, (**B**) Cu, (**C**) Mn.

**Figure 3 ijerph-18-12947-f003:**
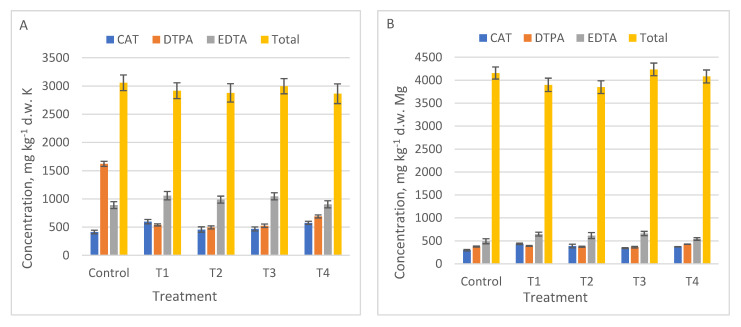
The mobile metallic concentration of macro nutrients in Control, T1 to T4 treatments using three single chemical extraction procedures: DTPA, CAT, EDTA compared to total content (average ± SD, *n* = 3): (**A**) K; (**B**) Mg, (**C**) Ca, (**D**) Fe.

**Figure 4 ijerph-18-12947-f004:**
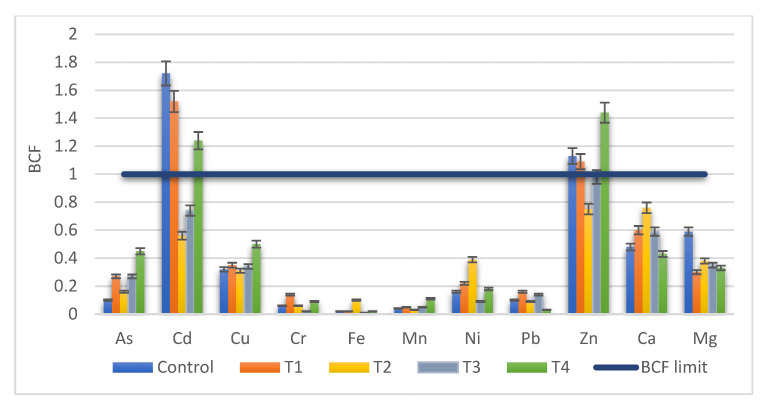
BCF of metals from soil to roots of *S. alba* in T1 to T4 treatments compared with Control and BCF limit = 1 (average ± SD, *n* = 3 for soil, *n* = 6 for plant).

**Figure 5 ijerph-18-12947-f005:**
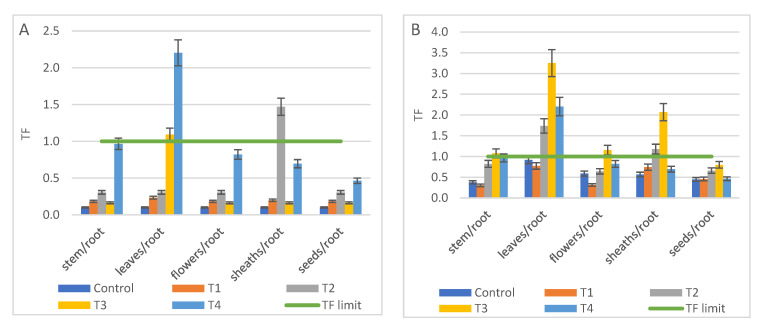
The translocation factor (TF) from root of *S. alba* to plant tissues in T1 to T4 treatments compared with control and TF limit = 1 (average ± SD, *n* = 6): (**A**) As, (**B**) Cd, (**C**) Ni.

**Figure 6 ijerph-18-12947-f006:**
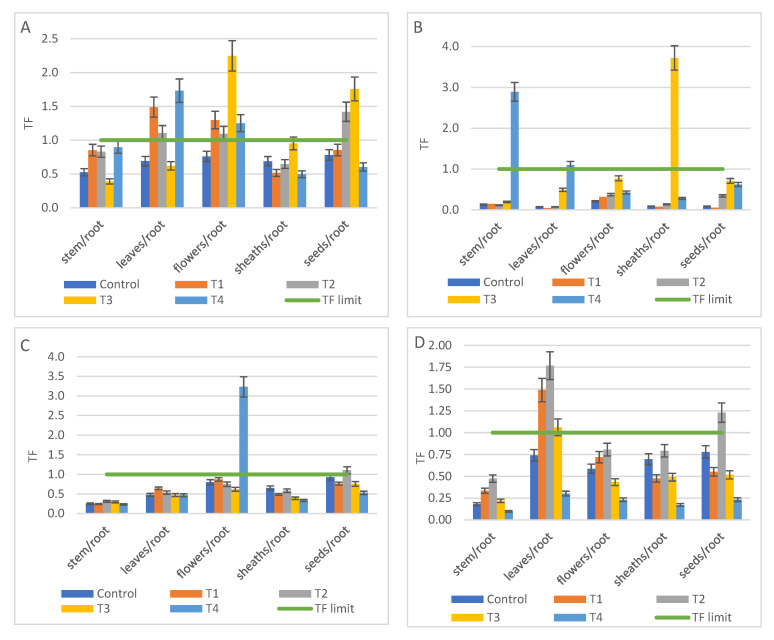
The translocation factor (TF) from root of *S. alba* to plant tissues in T1 to T4 treatments compared with control and TF limit (average ± SD, *n* = 6) for micronutrients: (**A**) Zn; (**B**) Cr; (**C**) Cu; (**D**) Mn.

**Figure 7 ijerph-18-12947-f007:**
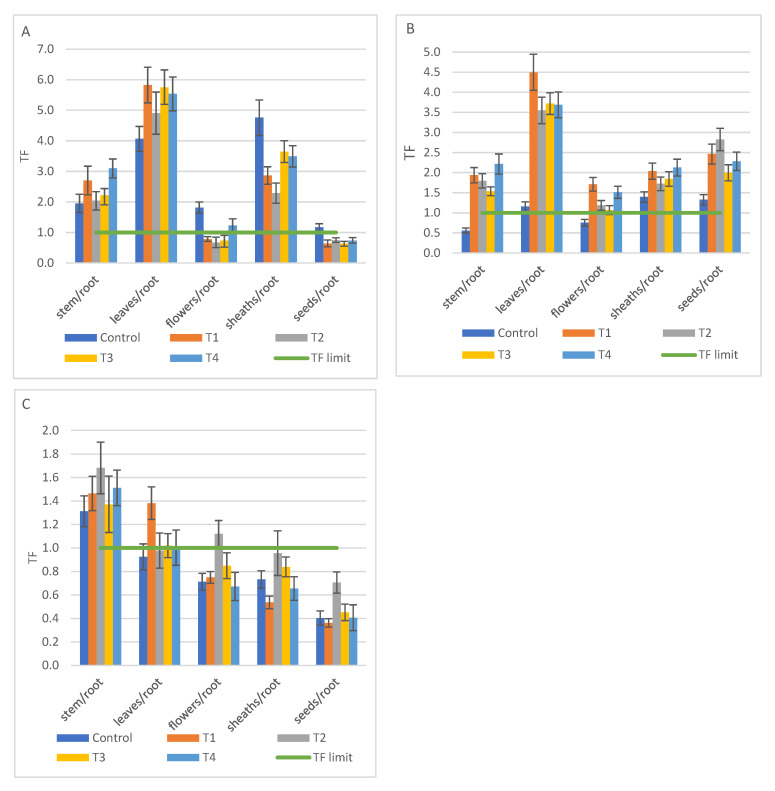
The translocation factor (TF) from root of *S. alba* to plant tissues in T1 to T4 treatments compared to control and TF limit (average ± SD, *n* = 6) for macro nutrients: (**A**) Ca; (**B**) Mg; (**C**) K.

**Table 1 ijerph-18-12947-t001:** Applied treatment.

Treatment	Treatment Code	Seedlings Plants/Test	Enrichment Concentration * (mg kg^−1^ d.w)			
			As	Cd	Ni	Pb
Control	C1, C2	5	-	-	-	-
As	T1-1, T1-2	5	15	-	-	-
As + Cd	T2-1, T2-2	5	15	3	-	-
As + Cd + Ni	T3-1, T3-2	5	15	3	140	-
As + Cd + Ni + Pb	T4-1, T4-2	5	15	3	70	70

Note: * planed nominal concentrations in the contaminated soils.

**Table 2 ijerph-18-12947-t002:** Romanian reference values for soils with sensitive uses (mg kg^−1^ d.w.) [42].

Metals	Normal Value	Alert Threshold	Intervention Threshold
As	5	15	25
Cd	1	3	5
Ni	20	75	150
Pb	20	50	100

**Table 3 ijerph-18-12947-t003:** Chemical conditions for bioavailable fraction extraction.

Code	Solution Mixtures	Extraction Conditions	Applied Standard
DTPA	0.005 mol L^−1^ DTPA + 0.1 mol L^−1^ TEA + 0.01 mol L^−1^ CaCl_2_	pH 7.3 ± 0.2, sol:solution ratio1:10, 2 h at 40 rpm min^−1^	ISO 14870/2001 [43]
CAT	0.01 mol L^−1^ CaCl_2_ + 0.002 mol L^−1^ DTPA	pH 2.6 ± 0.05, sol:solution ratio 1:5, 2 h at 40 rpm min^−1^	EN 13651/2001 [44]
EDTA	1 mol L^−1^ CH_3_COONH_4_ + 0.01 mol L^−1^ EDTA	pH 7.00 ± 0.02, sol:solution ratio 1:10, 2 h at 40 rpm min^−1^	NF X31-120/1992 [45]

**Table 4 ijerph-18-12947-t004:** Metal concentrations in mustard seeds, control and polluted soils, in mg kg^−1^ d.w. (average ± ue, *n* = 3).

Metals	Mustard Seeds	Control Soil	Soil-T1	Soil-T2	Soil-T3	Soil-T4
As	<0.75	<0.75	15.0 ± 2.30	15.4 ± 2.32	16.9 ± 2.52	15.7 ± 2.44
Cd	0.3 ± 0.03	0.1 ± 0.01	0.2 ± 0.03	3.1 ± 0.37	3.0 ± 0.36	2.8 ± 0.33
Cu	4.4 ± 0.53	26.9 ± 6.46	27.6 ± 6.62	26.0 ± 6.24	28.7 ± 6.89	27.9 ± 6.70
Cr	0.2 ± 0.02	16.1 ± 2.11	12.7 ± 1.70	11.9 ± 1.51	15.8 ± 2.11	13.7 ± 1.82
Fe	72.3 ± 5.82	19163 ± 1533	18169 ± 1454	19363 ± 1549	19818 ± 1585	18482 ± 1479
Mn	20.0 ± 1.62	530 ± 42	503 ± 40	493 ± 40	485 ± 39	511 ± 41
Ni	2.1 ± 0.25	16.9 ± 3.41	17.4 ± 3.52	15.2 ±3.04	134 ± 27	70.0 ± 14.11
Pb	<1.5	13.5 ± 2.03	13.1 ± 2.05	13.5 ± 2.03	13.4 ± 2.05	71.3 ± 11.21
Zn	52.4 ± 6.31	89.7 ± 10.82	83.7 ± 10.11	84.6 ± 10.21	74.3 ± 8.92	74.9 ± 9.04
Ca	4386 ± 526	11968 ± 1795	11185 ± 1678	10613 ± 1595	12458 ± 1869	11607 ± 1741
Mg	3014 ± 452	4155 ± 623	3898 ± 585	3848 ± 577	4234 ± 635	4081 ± 612
K	9625 ± 1444	3056 ± 458	2917 ± 438	2878 ± 432	2996 ± 449	2863 ± 429

Note: ue—expanded uncertainty; < value lower than the method quantification limit.

**Table 5 ijerph-18-12947-t005:** Statistical analysis of extraction methods used for mobile metals (*p*-values).

Metal.	CAT-DTPA	CAT-EDTA	DTPA-EDTA
	*p*-Value		
As	0.0145 *	0.0004 *	0.0863
Cd	0.4133	0.2521	0.1882
Ni	0.1568	0.0103 *	0.0007 *
Pb	0.1746	0.1862	0.4821
Cu	0.1991	0.3530	0.1121
Mn	0.1062	0.0050 *	0.0001 *
Zn	0.3940	0.0006 *	0.0012 *
Ca	2.6 × 10^−10^ *	0.0164 *	7.29 × 10^−7^ *
Mg	0.0061 *	0.0712	8.48 × 10^−5^ *
K	1.43 × 10^−7^ *	0.4710	1.85 × 10^−7^ *
Fe	0.0006 *	3.52 × 10^−5^ *	0.1829

Note: * *p*-value < 0.05 significant differences.

**Table 6 ijerph-18-12947-t006:** Toxic trace metals accumulation in plant tissues of *S. alba* (mean ± SD, *n* = 6), mg kg^−1^ d.w.

As					
Plant Tissue	Control	T 1	T 2	T 3	T 4
Root	0.75 ± 0.003 ^c^	4.11 ± 3.181 ^bc^	2.45 ± 0.262 ^bc^	16.08 ± 6.483 ^a^	7.00 ± 2.351 ^b^
Stem	0.75 ± 0.003 ^c^	0.96 ± 0.252 ^c^	0.75 ± 0.004 ^c^	4.99 ± 1.284 ^bc^	1.59 ± 0.592 ^c^
Leaves	0.75 ± 0.003 ^c^	0.82 ± 0.091 ^c^	3.60 ± 1.752 ^bc^	0.75 ± 0.002 ^c^	1.65 ± 0.934 ^c^
**Cd**					
Root	0.21 ± 0.161 ^ef^	-	1.76 ± 0.102 ^def^	2.19 ± 0.611 ^def^	3.45 ± 1.132 ^de^
Stem	0.08 ± 0.013 ^f^	-	1.45 ± 0.404 ^def^	2.36 ± 0.854 ^def^	2.6 ± 1.18 ^def^
Leaves	0.22 ± 0.052 ^ef^	-	3.05 ± 0.614 ^def^	7.87 ± 1.983 ^b^	7.1 ± 1.33 ^bc^
Flowers	0.11 ± 0.031 ^f^	-	1.13 ± 0.323 ^ef^	1.13 ± 0.322 ^ef^	24.4 ± 4.88 ^a^
Sheath	0.10 ± 0.042 ^f^	-	2.07 ± 0.283 ^def^	4.53 ± 1.051 ^cd^	2.4 ± 0.42 ^def^
Seeds	0.09 ± 0.013 ^f^	-	1.16 ± 0.102 ^ef^	1.75 ± 0.202 ^def^	1.26 ± 0.093 ^ef^
**Ni**					
Root	2.74 ± 1.131 ^g^	-	-	11.9 ± 2.39 ^def^	15.6 ± 4.86 ^cd^
Stem	0.33 ± 0.042 ^g^	-	-	4.6 ± 1.21 ^efg^	4.1± 0.17 ^fg^
Leaves	0.60 ± 0.544 ^g^	-	-	7.4 ± 2.47 ^defg^	11.3 ± 2.57 ^def^
Flowers	0.82 ± 0.603 ^g^	-	-	29.2 ± 9.95 ^b^	37.8 ± 3.90 ^a^
Sheath	0.49 ± 0.140 ^g^	-	-	11.3 ± 1.10 ^def^	7.4 ± 0.39 ^defg^
Seeds	0.29 ± 0.192 ^g^	-	-	20.9 ± 3.66 ^c^	12.8 ± 2.58 ^cde^

**Note**: Similar letters are statistically non-significant according to Tukey HSD Test (*p* < 0.05), data are means (*n* = 6) ± SD, *a* represents significantly highest value followed by later alphabet letters for lower means.

**Table 7 ijerph-18-12947-t007:** Micronutrients accumulation in plant tissues of *S. alba* (mean ± SD, *n* = 6), mg kg^−1^ d.w.

Zn					
Plant Tissue	Control	T 1	T 2	T 3	T 4
Root	138 ± 31.3 ^bcd^	91.5 ± 15.14 ^cdefghi^	63.6 ± 12.18 ^hi^	80.5 ± 18.38 ^defghi^	136 ± 33.9 ^bcdef^
Stem	53.2 ± 14.59 ^i^	80.4 ± 11.44 ^defghi^	52.8 ± 6.91 ^i^	66.0 ± 14.20 ^hi^	137 ± 28.5 ^bcde^
Leaves	77.5 ± 18.72 ^fghi^	142 ± 40.1 ^bc^	70.3 ± 12.94 ^ghi^	114.8 ± 55.52 ^bcdefgh^	211 ± 36.1 ^a^
Flowers	74.3 ± 25.57 ^ghi^	126± 9.4 ^bcdefg^	69.6 ± 15.78 ^ghi^	72.1 ± 11.40 ^ghi^	151 ± 23.8 ^b^
Sheath	70.3 ± 16.33 ^ghi^	45.9 ± 8.47 ^i^	44.8 ± 5.98 ^i^	67.9 ± 9.15 ^ghi^	59.1 ± 7.37 ^hi^
Seeds	79.1 ± 14.77 ^efghi^	89.2 ± 20.25 ^cdefghi^	94.9 ± 12.71 ^bcdefghi^	67.7 ± 12.98 ^ghi^	69.7 ± 8.74 ^ghi^
**Cu**					
Root	9.1 ± 1.15 ^cde^	10.0 ± 1.33 ^bc^	8.1 ± 1.52 ^cdef^	9.8 ± 4.25 ^bcd^	15.5 ± 3.11 ^b^
Stem	2.1 ± 0.21 ^f^	2.9 ± 1.13 ^ef^	2.5 ± 0.56 ^f^	2.9 ± 0.86 ^ef^	3.5 ± 0.81 ^def^
Leaves	4.6 ± 1.20 ^cdef^	6.5 ± 1.18 ^cdef^	4.3 ± 0.57 ^cdef^	4.6 ± 1.51 ^cdef^	6.8 ± 2.06 ^cdef^
Flowers	7.0 ± 2.74 ^cdef^	8.4 ± 0.95 ^cdef^	6.1 ± 0.94 ^cdef^	6.1 ± 0.94 ^cdef^	31.0 ± 9.47 ^a^
Sheath	5.6 ± 0.58 ^cdef^	4.4 ± 0.68 ^cdef^	4.7 ± 0.84 ^cdef^	3.9 ± 0.64 ^cdef^	5.2 ± 0.92 ^cdef^
Seeds	7.8 ± 2.02 ^cdef^	8.3 ± 1.49 ^cdef^	9.6 ± 2.09 ^bcd^	6.5 ± 1.79 ^cdef^	7.4 ± 1.18 ^cdef^
**Cr**					
Root	1.10 ± 0.192 ^ab^	2.08 ± 1.032 ^ab^	0.74 ± 0.162 ^b^	0.40 ± 0.192 ^b^	1.62 ± 0.822 ^ab^
Stem	0.17 ± 0.134 ^b^	0.30 ± 0.271 ^b^	0.10 ± 0.041 ^b^	0.07 ± 0.004 ^b^	2.96 ± 1.144 ^a^
Leaves	0.08 ± 0.031 ^b^	1.82 ± 3.463 ^ab^	0.07 ± 0.004 ^b^	0.19 ± 0.093 ^b^	1.22 ± 0.391 ^ab^
Flowers	0.23 ± 0.182 ^b^	0.64 ± 0.133 ^b^	0.29 ± 0.142 ^b^	0.23 ± 0.084 ^b^	1.69 ± 0.470 ^ab^
Sheath	0.09 ± 0.011 ^b^	0.18 ± 0.092 ^b^	0.12 ± 0.044 ^b^	1.59 ± 0.831 ^ab^	0.07 ± 0.003 ^b^
Seeds	0.09 ± 0.023 ^b^	0.11 ± 0.034 ^b^	0.30 ± 0.103 ^b^	0.22 ± 0.092 ^b^	0.29 ± 0.142 ^b^
**Mn**					
Root	21.6 ± 20.71 ^bcd^	21.4 ± 5.38 ^bcd^	13.0 ± 3.47 ^bcdefg^	17.5 ± 4.51 ^bcdefg^	21.2 ± 4.43 ^bcde^
Stem	3.9 ± 1.26 ^g^	7.7 ± 0.77 ^defg^	15.8 ± 3.71 ^bcdefg^	5.1 ± 1.41 ^fg^	7.3 ± 0.65 ^efg^
Leaves	18.9 ± 3.57 ^bcdef^	44.9 ± 11.52 ^a^	12.1 ± 1.83 ^bcdefg^	24.7 ± 6.28 ^b^	24.3 ± 2.5 ^bc^
Flowers	10.8 ± 3.44 ^bcdefg^	16.1 ± 1.95 ^bcdefg^	9.6 ± 4.02 ^defg^	10.1 ± 2.36 ^defg^	13.4 ± 1.32 ^bcdefg^
Sheath	15.1 ± 2.22 ^bcdefg^	12.5 ± 1.73 ^bcdefg^	10.5 ± 1.07 ^cdefg^	11.2 ± 1.32 ^bcdefg^	10.9 ± 1.72 ^bcdefg^
Seeds	16.9 ± 1.31 ^bcdefg^	12.6 ± 1.53 ^bcdefg^	13.3 ± 1.94 ^bcdefg^	12.1 ± 1.37 ^bcdefg^	14.2 ± 1.63 ^bcdefg^

**Note**: Similar letters are statistically non-significant according to Tukey HSD Test (*p* < 0.05), data are means (*n* = 6) ± SD, *a* represents significantly highest value followed by later alphabet letters for lower means.

**Table 8 ijerph-18-12947-t008:** Macro nutrients accumulation in plant tissues of *S. alba* (mean ± SD, *n* = 6), mg kg^−1^ d.w.

Ca					
Plant Tissue	Control	T 1	T 2	T 3	T 4
Root	5744 ± 1940 ^ij^	6176 ± 1354 ^ij^	5557 ± 343 ^ij^	4476 ± 208 ^j^	6412 ± 993 ^ij^
Stem	11202 ± 700 ^hig^	18227 ± 1369 ^ef^	13058 ± 1381 ^fgh^	16114 ± 2127 ^efg^	20164 ± 2347 ^de^
Leaves	26805 ± 5198 ^bc^	32146 ± 2901 ^b^	39539 ± 6378 ^a^	42433 ± 2757 ^a^	41541 ± 2195 ^a^
Flowers	6168 ± 1246 ^ij^	5240 ± 708 ^j^	5938 ± 370 ^ij^	4937 ± 659 ^j^	7644 ± 690 ^hij^
Sheath	27718 ± 2177 ^bc^	18598 ± 725 ^ef^	18306 ± 969 ^ef^	24765 ± 2383 ^cd^	25083 ± 2241 ^cd^
Seeds	6718 ± 348 ^ij^	4514 ± 785 ^j^	6027 ± 654 ^ij^	4698 ± 359 ^j^	4540 ± 318 ^j^
**Mg**					
Root	2203 ± 631 ^bcdefgh^	1163 ± 161 ^h^	1347 ± 164 ^gh^	1665± 11 ^efgh^	1494± 202 ^fgh^
Stem	1377 ± 157 ^gh^	2413 ± 302 ^bcdef^	2420 ± 134 ^bcdefg^	2547 ± 151 ^bcdef^	3137 ± 640 ^bcd^
Leaves	3309 ± 692 ^bc^	3222± 849 ^bc^	4782 ± 720 ^a^	5559 ± 1080 ^a^	5415 ± 780 ^a^
Flowers	1570± 221 ^efgh^	2010 ± 58 ^defgh^	1647 ± 127 ^efgh^	1646 ± 128 ^efgh^	2060 ± 168 ^defgh^
Sheath	3412 ± 91 ^b^	2682 ± 310 ^bcde^	2350 ± 199 ^bcdefg^	2453± 234 ^bcdefg^	3289 ± 201 ^cb^
Seeds	3267 ± 281 ^bc^	3002 ± 232 ^bcd^	3435± 286 ^b^	3262± 139 ^bc^	3034 ± 189 ^bcd^
**K**					
Root	33000 ± 5996 ^def^	40231 ± 1801 ^cd^	32096 ± 2188 ^defg^	31924 ± 2195 ^defg^	38938 ± 4599 ^cde^
Stem	43312 ± 7086 ^c^	53935 ± 5236 ^b^	38889 ± 2745 ^cde^	39090 ± 3585 ^cde^	46109 ± 6571 ^bc^
Leaves	30516 ± 7109 ^efgh^	67317 ± 3201 ^a^	21296 ± 2111 ^hijklmn^	25287 ± 2657 ^fghij^	33507 ± 3218 ^def^
Flowers	23494 ± 2891 ^ghijklm^	31270 ± 3311 ^defg^	27304 ± 1407 ^fghi^	28554 ± 1664 ^fghi^	24969 ± 2298 ^fghijk^
Sheath	13345 ± 563 ^n^	16601 ± 2362 ^jklmn^	15629 ± 1520 ^klmn^	14944 ± 2322 ^lmn^	14374 ± 1748 ^mn^
Seeds	33000 ± 5996 ^def^	40231 ± 1801 ^cd^	32096 ± 2188 ^defg^	31924 ± 2195 ^defg^	38938 ± 4599 ^cde^
**Fe**					
Root	515 ± 84 ^a^	482 ± 111 ^a^	235 ± 94 ^b^	263 ± 76 ^b^	334 ± 67 ^b^
Stem	21.0 ± 5.82 ^c^	19.6 ± 4.34 ^c^	20.8 ± 2.44 ^c^	17.6 ± 4.80 ^c^	30.1 ± 9.84 ^c^
Leaves	49.1 ± 9.71 ^c^	79.4 ± 14.82 ^c^	64.3 ± 10.32 ^c^	52.3 ± 6.81 ^c^	65.9 ± 7.82 ^c^
Flowers	72.9 ± 15.11 ^c^	93.0 ± 20.13 ^c^	63.9 ± 13.91 ^c^	64.9 ± 12.20 ^c^	58.4 ± 12.41 ^c^
Sheath	68.9 ± 7.63 ^c^	57.8 ± 17.21 ^c^	43.8 ± 9.83 ^c^	52.5 ± 3.52 ^c^	46.0 ± 7.10 ^c^
Seeds	53.5 ± 4.72 ^c^	73.6 ± 6.32 ^c^	74 ± 11 ^c^	74.5 ± 4.04 ^c^	65.3 ± 4.82 ^c^

**Note**: Similar letters are statistically non-significant according to Tukey HSD Test (*p* < 0.05), data are means (*n* = 6) ± SD, *a* represents significantly highest value followed by later alphabet letters for lower means.

## Data Availability

The data are not publicly available due to ongoing research in this field carried out within the below mentioned project.

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
