# Peer review of "Bioavailability, Accumulation and Distribution of Toxic Metals (As, Cd, Ni and Pb) and Their Impact on Sinapis alba Plant Nutrient Metabolism"

_ijerph, 2021, doi:10.3390/ijerph182412947_

Round 1

Reviewer 1 Report

The manuscript of Vasile et al. deals with an interesting and important topic. The experiment is well planned and also methods assure a good reliability of results. However, some points must be carefully revised:

  1. The major concern is about statistical analyses. It has no sense to present all data and after report again just a part of results with statistical analyses. Authors must perform statistics for all their data and present the results immediately in the figures. I strongly recommend to revise this part.
  2. The discussion seems just a summary of results. Please rewrite this section including a critical analyses of results obtained in this work and results present in literature, referring also to the introduction.
  3.  

    Please use the word treatment instead of experiment to define the different conditions that you applied on plants. The terms is really confusing. It would be better to modify also the treatment abbreviations in T1, T2, T3, ..., rather than E1, E2... Avoid also the use of 'test'.
  4. The number of replicates is not clear, in all figure captions 3 replicates are reported while in table 1 two replicates of 5 plants are indicated. How many replicates have you used? For BCA, 6 replicates are reported, are these technical replicates of biological replicates? 
  5. Abbreviations must be always use along the manuscript. Also for metal names.
  6. Always use past tense for results description. The alternate use of present and past tenses occurs along all the manuscript. I tried to highlight some points but it occurs really too frequently.
  7. I reported several minor comments and typos in the attached file.

Author Response

Dear anonymous reviewer,

Thank you for the suggestions that have led to the considerable improvement of the manuscript. Please, find below the answer to your suggestion for improving the manuscript:

  1. The major concern is about statistical analyses. It has no sense to present all data and after report again just a part of results with statistical analyses. Authors must perform statistics for all their data and present the results immediately in the figures. I strongly recommend to revise this part.

In order to answer to your suggestion, we re-organize all the data and we selected the same trace metals in entire paper (As, Cd, Ni, Pb, Cu, Cr, Mn, Zn, Ca, Mg, K, Fe). Also, we divided in three groups: toxic metals (As, Cd, Ni, Pb), micro nutrients (Cr, Cu, Mn, Zn) and macro nutrients (Ca, Mg, K, Fe). We provided the statistical analyses in plant tissues for all selected metals. We deleted figure 4 (Accumulation of metals in different tissues of S. Alba) and figures 8 to 13 (As, Cd, Ni, Cu, Cr, Zn – concentrations in all tissues of white mustard) and replaced with tables 5 to 7, where are presented the concentrations in plant tissues – statistical analysis. We chose to exclude the graphs because they presented the same data, in the first case ordered by the plant tissue, and in the second case after the treatment. We add comments to the section: Metal concentration in plant tissue after exposure. Statistical analyses.

In addition, we introduced graphs and comments for macronutrient translocation (Ca, Mg, K) and gave information about Fe translocation (figure 8A – 8C).

We have added a chart for mobile Fe (figure 3D) and given information about Cr mobility.

We introduced supplementary information in comments to figure 5 regarding BCF of K in all treatments.

We removed from table 4 the data regarding Co and Na, which are not found in the data presented in the plants.

  1. The discussion seems just a summary of results. Please rewrite this section including a critical analyses of results obtained in this work and results present in literature, referring also to the introduction.

The introduction and the discussion chapter have been reorganized and completed.

  1. Please use the word treatment instead of experiment to define the different conditions that you applied on plants. The terms is really confusing. It would be better to modify also the treatment abbreviations in T1, T2, T3, ..., rather than E1, E2... Avoid also the use of 'test'.

We have replaced in the manuscript “experiment” /” test” with treatment, also we replaced E1, E2...E4 with T1, T2...T4 in all figures and tables.

  1. The number of replicates is not clear, in all figure captions 3 replicates are reported while in table 1 two replicates of 5 plants are indicated. How many replicates have you used? For BCA, 6 replicates are reported, are these technical replicates of biological replicates?

We have clarified in the text the number of replicas used. Thus, due to its homogeneity, the soil was analyzed both for total content and for mobile fraction in three different samples (soil, n=3) before being added to the two test containers. Regarding the plants, three specimens of each replicate were harvested when they reached flowering, separated into organs and analyzed. Thus, six different plants were analyzed for the same treatment (plants, n=6). The other 2 plants that remained in each pot for a specific treatment, were left to reach maturity for seeds harvesting. In this case, the seeds were harvested and 3 different samples were analyzed for each pot, a total of 6 seed samples per treatment.

  1. Abbreviations must be always use along the manuscript. Also for metal names.

We have modified and introduced abbreviations for both metals and other compounds mentioned in the text, such as: P, N, TOC.

  1. Always use past tense for results description. The alternate use of present and past tenses occurs along all the manuscript. I tried to highlight some points but it occurs really too frequently.

We have corrected the form of the verbs throughout the manuscript.

  1. I reported several minor comments and typos in the attached file.

Regarding the comments in the attached file, we made corrections, such as:

  • mg/kg d.w. or g/L, mol/L: mgkg-1w., gL-1, molL-1. In text, figures and tables;
  • we changed the title and keywords;
  • we made all the suggested corrections

Please see more corrections in the updated manuscript.

Reviewer 2 Report

The study by Vasile et al. not only investigated the bioavailability, accumulation and distribution of Ni, As, Cd & Pb, but also the levels of selected micro- and macronutrients in the soil and plant (Sinapis alba) tissues. Kindly, I have some comments that I request the authors to address before this article can be considered for publication.

Even though Arsenic is generally included in the group termed as ‘heavy metals’ because of its lack of biological functions and toxicity, the element is a metalloid. I propose that the authors should use the term ‘trace elements’ when referring to As, Cd, Ni and Pb in the same phrase or sentence. Please, all scientific names of the plants mentioned in the text should be in italics.

Since the current study was not just an investigation of the bioavailability, accumulation and distribution of Ni, As, Cd & Pb, but also the levels of selected micro- and macronutrients, the authors should consider modifying the title and the abstract to reflect the comprehensive nature of the study.

Kindly, I urge the authors to make a few changes in the introduction: a clear communication of the motivation and rationale for the study is needed. Why was this study done? Are there particular gaps that the authors intended to bridge? Briefly mention the rationale for selecting Sinapis alba. Is it a medicinal plant, vegetable, fodder etc? Is it because there is paucity of data regarding trace elements accumulation in this plant? If so, then these should be clearly mentioned in the introduction. There are several short paragraphs that can be merged to enhance continuity of similar ideas. For example: paragraphs 3,4 & 5, and 6 & 7. Paragraphs 8-12 can be re-written into two paragraphs: one for the essential elements e.g. (Ni & Zn) and the other for the non-essential elements (As, Cd and Pb). Kindly, correct the typos on the first line of the last paragraph ‘This study prsents the evolution…’. I presume that you meant ‘This study presents the evaluation…’

The methods, results and discussions are sound; unfortunately, there are numerous typos and grammatical mistakes that should be corrected. Kindly, avoid re-writing the results in the discussion section: they are already available in the results text, figures and tables. In the discussion, I missed the possible explanations for preferential accumulation of As and Cd in the sheaths and leaves, as opposed to the seeds.

The conclusion is unnecessarily long, with the first four paragraphs largely a repetition of the results.

Additional comments are available on the marked-up version of the manuscript attached.

Author Response

Dear anonymous reviewer,

Thank you for the suggestions that have led to the considerable improvement of the manuscript. Please, find below the answer to your suggestion for improving the manuscript:

 1. Even though Arsenic is generally included in the group termed as ‘heavy metals’ because of its lack of biological functions and toxicity, the element is a metalloid. I propose that the authors should use the term ‘trace elements’ when referring to As, Cd, Ni and Pb in the same phrase or sentence. Please, all scientific names of the plants mentioned in the text should be in italics.

We changed in the manuscript term “heavy metals” with “trace elements” regarding As, Cd, Ni and Pb.

We have changed the scientific name of the plants in italics.

  1. Since the current study was not just an investigation of the bioavailability, accumulation and distribution of Ni, As, Cd & Pb, but also the levels of selected micro- and macronutrients, the authors should consider modifying the title and the abstract to reflect the comprehensive nature of the study.

We followed the suggestion made and modified the title, we filled in the abstract In addition, we provided in the manuscript additional data on both macro nutrients and micro nutrients. Your suggestion helped us to restructure the material so that the metals were divided into toxic metals (As, Cd, Ni, Pb), micro nutrients (Cr, Cu, Mn, Zn) and macro nutrients (Ca, Mg, K, Fe). We added additional data in the form of graphics, tables or text comments to highlight the plant's behavior in terms of toxic metals, micro and macro nutrients.

  1. Kindly, I urge the authors to make a few changes in the introduction: a clear communication of the motivation and rationale for the study is needed. Why was this study done? Are there particular gaps that the authors intended to bridge? Briefly mention the rationale for selecting Sinapis alba. Is it a medicinal plant, vegetable, fodder etc? Is it because there is paucity of data regarding trace elements accumulation in this plant? If so, then these should be clearly mentioned in the introduction. There are several short paragraphs that can be merged to enhance continuity of similar ideas. For example: paragraphs 3,4 & 5, and 6 & 7. Paragraphs 8-12 can be re-written into two paragraphs: one for the essential elements e.g. (Ni & Zn) and the other for the non-essential elements (As, Cd and Pb). Kindly, correct the typos on the first line of the last paragraph ‘This study prsents the evolution…’. I presume that you meant ‘This study presents the evaluation…’

In order to improve discussion section, the introduction was also improved, to make the connection between the state of art of the subject and the results obtained in the study. The entire section has been rewritten, new comments and references have been added.

The paragraphs mentioned above have been corrected and merged.

  1. The methods, results and discussions are sound; unfortunately, there are numerous typos and grammatical mistakes that should be corrected. Kindly, avoid re-writing the results in the discussion section: they are already available in the results text, figures and tables. In the discussion, I missed the possible explanations for preferential accumulation of As and Cd in the sheaths and leaves, as opposed to the seeds.

The English has been corrected.

  1. The conclusion is unnecessarily long, with the first four paragraphs largely a repetition of the results.

The conclusions were shortened.

Round 2

Reviewer 1 Report

I appreciate authors' efforts in improving the manuscript. I have some small comments as follows.

Abstract must be shortened.

Please space the units of measure, mgkg-1 is mg kg-1

In introduction section, please add some information regarding the effect of Pb on plant physiology for consistency with other metal descriptions.

Wikipedia cannot be used as reference in a research paper, please refer to a WoS journal.

Please explain why a different Ni concentration has been used in the two treatment with Ni.

Please remove a decimal number in average values in table or add a decimal number in UE/SE values.

Maybe it is a formatting issue, but 5 panels are reported in Figure 2 rather than 3 while panel D in figure 3 does not have total Fe concentration.

Please add statistics also for BCA and TF values as well as for the metal concentrations obtained using different extraction methods. Without statistical analyses authors cannot state if differences between the extraction methods are real.

BCA and TF are measures to clarify the plant attitude to accumulate toxic compounds and thus its potentiality for phytoremediation approaches. In my opinion this calculation has no sense for essential micro and macro nutrients.

Author Response

Thank you for your time and the suggestions and comments. Please see in the attachment our answer.
